# Failed Disruption of Tick Feeding, Viability, and Molting after Immunization of Mice and Sheep with Recombinant *Ixodes ricinus* Salivary Proteins IrSPI and IrLip1

**DOI:** 10.3390/vaccines8030475

**Published:** 2020-08-26

**Authors:** Consuelo Almazán, Lisa Fourniol, Sabine Rakotobe, Ladislav Šimo, Jérémie Bornères, Martine Cote, Sandy Peltier, Jennifer Maye, Nicolas Versillé, Jennifer Richardson, Sarah I. Bonnet

**Affiliations:** 1UMR BIPAR 0956, INRAE, National Veterinary School of Alfort, ANSES, Paris-Est University, 94700 Maisons-Alfort, France; c_almazan_g@hotmail.com (C.A.); fourniol-lisa@hotmail.fr (L.F.); sabine.rakotobe@anses.fr (S.R.); ladislav.simo@vet-alfort.fr (L.Š.); martine.cote94700@gmail.com (M.C.); 2SEPPIC Paris La Défense, 92250 La Garenne Colombes, France; jeremie.borneres@airliquide.com (J.B.); sandy.peltier@inserm.fr (S.P.); jennifer.maye@airliquide.com (J.M.); Nicolas.VERSILLE@airliquide.com (N.V.); 3UMR Virologie 1161, INRAE, National Veterinary School of Alfort, ANSES, Paris-Est University, 94700 Maisons-Alfort, France; jennifer.richardson@vet-alfort.fr

**Keywords:** ticks, anti-tick vaccination, Ixodes ricinus, salivary proteins

## Abstract

To identify potential vaccine candidates against *Ixodes ricinus* and tick-borne pathogen transmission, we have previously sequenced the salivary gland transcriptomes of female ticks infected or not with *Bartonella henselae.* The hypothesized potential of both IrSPI (*I. ricinus* serine protease inhibitor) and IrLip1 (*I. ricinus* lipocalin 1) as protective antigens decreasing tick feeding and/or the transmission of tick-borne pathogens was based on their presumed involvement in dampening the host immune response to tick feeding. Vaccine endpoints included tick larval and nymphal mortality, feeding, and molting in mice and sheep. Whether the antigens were administered individually or in combination, the vaccination of mice or sheep elicited a potent antigen-specific antibody response. However, and contrary to our expectations, vaccination failed to afford protection against the infestation of mice and sheep by *I. ricinus* nymphs and larvae, respectively. Rather, vaccination with IrSPI and IrLip1 appeared to enhance tick engorgement and molting and decrease tick mortality. To the best of our knowledge, these observations represent the first report of induction of vaccine-mediated enhancement in relation to anti-tick vaccination.

## 1. Introduction

Worldwide, vector-borne diseases (VBD) account for more than 17% of all infectious diseases in humans, causing 1 billion cases and 1 million deaths per year [1]. In addition, VBD outbreaks in domestic animals and wildlife disrupt ecosystems, impose a significant burden on animal health, and are an impediment to socioeconomic development [2]. Among arthropods responsible for VBD, ticks are the most important vectors of viral, bacterial, and parasitic pathogens that affect animals worldwide, and they are second only to mosquitoes where humans are concerned [3]. In Europe, *Ixodes ricinus* is the most widespread and abundant tick, and it is the vector of several tick-borne diseases (TBD) of medical and veterinary importance, including Lyme borreliosis, anaplasmosis, rickettsiosis, babesiosis, and tick-borne encephalitis (TBE) in humans, and babesiosis and anaplasmosis in livestock [4]. In Europeans, the most prevalent TBD is Lyme borreliosis, with an estimated 85,000 cases each year [5], but many other zoonotic pathogens can be acquired by tick bites. In this last regard, recent transcriptomic studies using next-generation sequencing (NGS) techniques have led to the identification of several unexpected bacteria, viruses, and parasites in *I. ricinus* ticks from Eastern France, some of them representing potential pathogens for humans or animals [6,7,8]. To date, the intensification of human and animal movements and socioeconomic and environmental changes have led to the redistribution of certain tick species—that is, an extension of seasonal transmission periods and geographical distribution, as well as the appearance of TBD in previously unaffected areas, highlighting the urgent need to find better methods of control [9,10,11,12].

Current tick control strategies essentially rely on the use of chemical acaricides and repellents. However, their widespread deployment has led to the selection of resistance in multiple species of ticks [13]. Moreover, these products are responsible for environmental contamination and, in farm animals, the contamination of milk and meat products with drug residues [14]. Thus, new approaches that are environmentally sustainable and that provide broad protection against current and future tick-borne pathogens (TBP) are urgently needed. In light of the limited understanding of immunity to TBP, TBP strain diversity and, more generally, the transmission of multiple TBP by the same tick species, vaccine strategies targeting conserved tick molecules that play key roles in tick biology and/or vector competence are increasingly being sought. Indeed, immunity to such molecules holds the promise of affording broad protection against multiple TBD [15,16]. In this endeavor, the primary rate-limiting step is the identification of protective antigenic targets [17].

The use of tick antigens for vaccinal purposes was first documented in 1939 and involved the immunization of guinea pigs with extracts of *Dermacentor variabilis* [18]. The vaccine Gavac^TM^ (Heber Biotec S.A., Havana, Cuba), which is based on the Bm86 antigen, a midgut protein of *Rhipicephalus microplus*, has been used in several Latin American countries since 1993 and is currently the only ectoparasite vaccine that is commercially available [19]. Antibodies elicited against Bm86 are believed to lyse the tick’s gut wall, thus interfering with feeding and subsequent egg production. When applied within an integrated management strategy, such vaccines are effective in reducing tick burden and hence the transmission of certain TBP [20]. However, for vaccines such as these, whose impact on TBD is secondary to their effect on tick burden, a reduction in TBD is unlikely to be achieved unless the targeted tick species feeds principally on the host for which the vaccine is intended. Such holds true for the so-called cattle tick *R. microplus,* which—while capable of feeding on a variety of hosts—mainly infests cattle in areas where they are present. However, such is not the case for many species of ticks responsible for important TBD, and notably for *Ixodes* sp., which feed relatively indiscriminately on multiple hosts and notably on wildlife. For these ticks, vaccines that interrupt tick feeding prior to pathogen transmission or that directly suppress vector competence must be sought.

With this aim in view, salivary antigens represent attractive vaccine candidates, as their neutralization by immune effectors might interfere with completion of the blood meal and subsequently with pathogen transmission, and because exposure to ticks could maintain immunity in vaccinated hosts [21]. Of note, many proteins present in tick saliva dampen host defenses to assure adequate feeding, thereby creating a favorable context for the survival and propagation of TBP (reviewed in [22]). However, vaccine-elicited antibodies against at least some tick antigens have proven to afford protection against natural challenge, despite possible interference by the immunosuppressive activities of tick saliva. Encouraging results have been actually obtained for several tick species (reviewed in [23]), including *I. ricinus*, for which several salivary gland antigens have been identified as potential vaccine candidates. These include metalloproteases and a serpin (iris) whose neutralization disrupts the feeding process [24,25], as well as the recombinant tick cement protein, 64TRP, that affords protection against TBE transmission to mice [26]. Recently, the use of a combination of transcriptomic and proteomic data has led to the identification of several antigens of interest, which included an uncharacterized secreted *I. ricinus* antigen that provided greater than 80% protection against tick infestation in rabbits and dogs [27]. In order to identify genes involved in either tick feeding or the vector competence of *I. ricinus*, we have previously identified transcripts induced in female tick salivary gland (SG) tissue in response to colonization by *Bartonella henselae* [28]. Although the majority of human cases are due to scratches and bites from infected cats, we have previously demonstrated that the bacteria may also be transmitted by *I. ricinus* [29]. Two of the identified genes—*IrSPI* (*Ixodes ricinus* serine protease inhibitor) and *IrLip1* (*Ixodes ricinus* lipocalin 1)—have been selected for evaluation as vaccine candidates against tick infestation and pathogen transmission.

*IrSPI* was selected due to being the most overexpressed gene following bacterial infection, encoding a predicted secreted protein, and belonging to the serine protease inhibitor family whose members assure diverse functions in ticks, including blood digestion, innate immunity, reproduction, and pathogen transmission [30]. We first demonstrated that *IrSPI* silencing both impaired tick feeding and reduced bacterial load in tick SGs [28]. In a recent study, we also showed that IrSPI is a Kunitz elastase inhibitor that is overexpressed in several tick organs—especially SG—during blood feeding, and that is injected into the host through saliva. While having no impact on tissue factor pathway-induced coagulation, fibrinolysis, endothelial cell angiogenesis, or apoptosis, the protein exhibits immunomodulatory activity, repressing the proliferation of CD4^+^ T lymphocytes and pro-inflammatory cytokine secretion from both splenocytes and macrophages [31].

*IrLip1* was not up-regulated in the tick SG following infection, but it was selected as encoding a putative secreted protein that belongs to the lipocalin family of proteins implicated, among other functions, in the inflammatory response of the vertebrate host [32]. Lipocalins are small extracellular proteins that bind to histamine, serotonin, and prostaglandin. In ticks, they are injected into the host within saliva, and they have been demonstrated to be implicated in the modulation of the host immune response, the regulation of hemostasis, and the clearance of endogenous and exogenous compounds [33]. Beaufays and co-workers identified 14 lipocalins from *I. ricinus*, among which LIR6 was shown to inhibit the classical or alternative complement pathway, allowing efficient blood meal acquisition [34]. In addition, a recent study of Contreras and co-workers reported the identification of a lipocalin in *I. scapularis* that has been proposed as a potential vaccine candidate against both tick infestation and *Anaplasma phagocytophilum* transmission [35].

In the present work, recombinant IrSPI and IrLip1 proteins were evaluated as vaccine candidates against both *I. ricinus* infestation in mouse and sheep models, and against *A. phagocytophilum* transmission to sheep though tick bite. *A. phagocytophilum* is an obligate intracellular Gram-negative bacterium transmitted by I. ricinus in Europe [36] and is the causative agent of human granulocytic anaplasmosis (HGA) and tick-borne fever (TBF), affecting both humans and a variety of domestic and wild animal species [37]. Unfortunately, the impact of vaccination on bacterial transmission could not be assessed here, owing to an inadequate engorgement of *A. phagocytophilum*-infected nymphs on sheep, as summarized below.

## 2. Material and Methods

### 2.1. Ethics

This study was carried out in strict accordance with the recommendation of the European Guide for the Care and Use of Laboratory Animals [38]. The protocols involving mice and sheep were approved by the local ethics committee for animal experimentation, ComEth Anses/ENVA/UPEC, (Permit numbers 20150914113472401 and 2016092716395004, respectively).

### 2.2. Ticks

*I. ricinus* larvae and nymphs used in these experiments were obtained from our pathogen-free colony maintained at the tick rearing facility of ANSES (The French Agency for Food, Environmental and Occupational Health & Safety, Maisons-Alfort, France) and reared at 22 °C with 95% relative humidity and a 12 h light/dark cycle, as previously described by Bonnet et al. [39].

### 2.3. Identification and Production of Recombinant Proteins

Both IrSPI (GenBank accession number: KF531922.2) and IrLip1 (GenBank accession number: MT133882) were identified as previously described in Liu et al. [28]. The two proteins corresponded to transcripts identified in salivary glands of pre-fed female *I. ricinus* that were infected or not at the larval and nymphal stage with *B. henselae*.

The two proteins were expressed in *Drosophila* S2 cells (Invitrogen, Carlsbad, CA, USA) at the Recombinant Proteins in Eukaryotic Cells Platform of Institut Pasteur. Paris, France. Both recombinant proteins were expressed and purified as previously described for IrSPI [31]. Briefly, the open reading frames (ORF) of IrSPI and IrLip1 lacking the 5′ sequence encoding their signal peptide (MKATLVAICFFAAVSYSMG and MGLQYALLFACVAAEEVWA, respectively) were synthetized (Eurofins Scientific) as fusion proteins with sequences encoding Twin-Strep-Tag and enterokinase. After insertion into the pMT/BIP/V5-His plasmid (ThermoFisher, Waltham, MA, USA), proteins were produced in *Drosophila* S2 cells (Invitrogen, Carlsbad, CA, USA) co-transfected with pMT/BIP/IrSPI or pMT/BIP/IrLip1 and the pCoPURO vector (Addgene, Teddington, UK) conferring resistance to puromycin, using the Cellfectin II transfection reagent (Invitrogen). The upstream pMT/BiP/V5-His *Drosophila* BiP secretion signal enables the recombinant protein to enter the S2 cell secretory pathway for recovery in culture medium. Then, protein expression was driven by the metallothionein promoter after induction by 5 µM CdCl_2_. Clarified cell supernatants were concentrated 10-fold using Kicklab tangential flow filtration cassettes (GE Healthcare, Boston, USA, cut-off 10 kDa) and adjusted to pH 8.0 in 10 mM Tris before purification in an AKTA Avant system by Steptrap-HP chromatography (GE Healthcare). Final purification was achieved by gel filtration using a Superdex 75 HiLoad 16/60 column (GE Healthcare) equilibrated in PBS-X (Phosphate Buffer Saline without Ca^2+^ and Mg^2+^). Protein quantity was estimated by peak integration.

### 2.4. Vaccine Formulation

Recombinant proteins were concentrated by the Amicon Ultra−4 ultracentrifugation system (Millipore-Merc, Darmstadt, Germany). Recombinant antigens or saline control (phosphate-buffered saline, PBS) were adjuvanted in Montanide ™ ISA 201 VG (Seppic, La défence, France) 24 h before immunization and kept at 4 °C until use. For mice, 0.1 mL doses of 10 µg of each recombinant protein were used per animal, while 1 mL doses of 50 µg of each recombinant protein were used for sheep, including for the combination of the two proteins.

### 2.5. Mice Immunization and Infestation

The vaccination and tick infestation of mice were performed at the facilities of the Biomedical Research Center (CRBM), National Veterinary School of Alfort (ENVA). Three groups of 5 female BALB/c mice, 8 weeks of age, were used. Mice from groups 1 and 2 were vaccinated with recombinant IrSPI and IrLip1, respectively, and mice from group 3 received only adjuvant emulsified with PBS and served as control. All of the mice were immunized subcutaneously in the abdomen region at days 0, 14, and 28 using 1 mL syringes with 27G needles in a previous disinfected area. Fourteen days after the last immunization (Day 42 post injection), each mouse was infested with 20 *I. ricinus* nymphs deposited in plastic capsules glued to the mouse back as previously described [40]. Engorged tick nymphs were collected daily after 3 days of infestation until day 7 when all of the ticks dropped off. Ticks were weighed and incubated at 22 °C and 80% humidity until molting. The tick nymphs that successfully molted to adult stage were collected and counted after 57 days of incubation. Tick engorgement (No. of engorged nymphs/No. of nymphs used for infestation), mortality (No. of dead nymphs at day 57/total No. engorged nymphs), feeding (mean weight after feeding/total No. engorged nymphs) and molting (No. adults at day 57/No. engorged nymphs) were evaluated and compared between vaccinated mice and controls. The statistical significance of the vaccination results shown were examined by student’s *t*-test with unequal variance (*p* = 0.05) and the *X*^2^ test (*p* = 0.05). A correlation analysis was conducted in Microsoft Excel (version 16.16.19, Microsoft, Redmond, DC, USA) to compare the effect of vaccination on engorgement, mortality, feeding, and molting after feeding on vaccinated or control mice with antibody titers at Day 42.

### 2.6. Sheep Immunization and Infestation

Vaccination and tick infestation were performed at the Infectiology Facility Val de Loire, INRAE Nouzilly, France. Four groups of six 7-month-old male sheep (PreAlps breed) were immunized intramuscularly, as previously reported by Hope et al. [41], with three doses of IrSPI (group 1), IrLip1 (group 2), a combination of IrSPI and IrLip1 (group 3), or with adjuvant emulsified with PBS (control group). Each animal was vaccinated using 5 mL syringes with 16-gauge needles at days 0, 15, and 30. Fifteen days after the last immunization, all of the animals were moved to a biosafety level 2 enclosure where the backs of the sheep were shaved, and cotton bags were placed on both sides of the back as previously described [42]. After 24 h, each sheep was infested with approximately 1000 larvae and 24 h later, each sheep was infested with 48 nymphs that had engorged as larvae on a sheep infected with *A. phagocytophilum* (NV2Os strain), resulting in an estimated infection rate in nymphs of 42%, as previously described [42]. After 24 h of infestation, all of the unattached ticks were removed from the cells and counted. Fully engorged ticks that dropped off into the cells were collected two times per day over 3 days of feeding for larvae and 4 days of feeding for nymphs. After every collection, ticks were washed, counted, and incubated at 22 °C and 80% of humidity until molting. The tick larvae and nymphs that successfully molted to the nymph and adult stages, respectively, were collected and counted after 90 days of incubation. For both larvae and nymphs, tick engorgement (number of engorged ticks/number of ticks used for infestation), mortality (number of dead ticks at day 90/total number engorged-attached ticks), and molting (number of adult nymphs at day 90/number of alive engorged larvae-nymphs) were evaluated and compared between vaccinated sheep and controls. The statistical significance of the vaccination results shown were examined by the *X*^2^ test (*p* = 0.05). A correlation analysis was conducted in Microsoft Excel (version 16.16.19) to compare the effect of vaccination on tick engorgement, mortality, and molting after feeding on vaccinated or control sheep with antibody titers at Day 45.

### 2.7. Clinical Follow-Up of Sheep Vaccination and Infection with Anaplasma phagocytophilum

To evaluate any local reaction at the site of the vaccine injection, changes in color and inflammation were monitored every 24 h for three days after each injection in both mice and sheep [43]. Sheep were vaccinated within a shaved area, and a circle was traced around the point of injection. In order to evaluate the innocuity of the vaccination and infection of sheep with *A. phagocytophilum* after tick infestation, clinical signs were also monitored, and for sheep, temperature was followed throughout the experiment using an intraruminal thermobolus system (Medria, Chateaugiron, France) that was orally administered seven days before the first immunization. The temperature was recorded per hour, and the daily average was calculated and plotted against each day, starting from the day of tick infestation (day 45) and lasting until day 54.

PCR detection of *A. phagocytophilum* in sheep was performed using DNA obtained from blood collected from the jugular vein 1 day before tick infestation, and at days 5, 10, and 15 afterwards, and following the protocol described by Kocan and co-workers for msp4 gene detection [44].

### 2.8. Determination of Antibody Levels in Serum from Immunized Animals by Enzyme Linked Immunosorbent Assay (Elisa)

Mouse blood samples were collected from the retro-orbital sinus with high precision Pasteur pipettes before each immunization, the day of tick infestation (day 42), at day 56, and at the end of the experiment (day 90). After centrifugation, serum was collected and stored at −20 °C until use.

Sheep blood samples were collected from the jugular vein in a previously disinfected area using 10 mL vacutainer tubes. Samples were obtained before each immunization (days 0, 15, 30), at tick infestation (day 45), at day 51, and at the time of euthanasia (day 73). Sera were collected following centrifugation and stored at −20 °C until ELISAs were performed.

Polystyrene microtiter ELISA plates (MaxiSorb, NUNC, Roskilde, Danemark) were coated with 100 µL per well of 10 µg/mL of recombinant proteins diluted in carbonate–bicarbonate buffer, pH 9.6 (Sigma-Aldrich, Darmstadt, Germany). The plates were incubated at 37 °C for two hours and washed three times with 250 µL of PBS containing 0.05% Tween 20 (Montanox 20, Seppic, France). After all incubations, the plates were washed three times with 250 µL of PBS containing 0.05% Tween 20 (Montanox 20, Seppic, France). The plates were blocked with PBS containing 1% bovine gelatin (Sigma) overnight at room Please state manufacturer, city and country from where equipment has been sourced. Please state the city and country that the company is located in.temperature. Serial dilutions of serum were added in duplicate. The plates were incubated with 1:3000 rabbit anti-sheep immunoglobulin G (IgG)–horseradish peroxidase conjugate (Thermo Fisher Scientific, Waltham, MA, USA) and 1:6000 goat anti-mouse immunoglobulin G1 (IgG1)–horseradish peroxidase conjugate (Thermo Fisher Scientific,) for the detection of sheep and mouse antibodies, respectively. The reaction was revealed by adding 3,3′,5,5′-tetramethylbenzidine (Fisher) for 5 min and stopped with 50 µL of 2N H_2_SO_4_. The optical density (450 nm) was measured in an ELISA reader (ThermoScientific). The antibody titers were expressed in arbitrary units (AU) by reference to a calibrated standard anti-ovalbumin IgG.

## 3. Results

### 3.1. Responses to Vaccination

Although body temperature was not monitored in mice, no clinical signs were observed after vaccination, and all of the animals remained healthy until the end of the experiment. For sheep, no clinical signs, such as a rise in temperature within 20 days of the last vaccination (Figure 1), or a local reaction at the site of injection, were observed.

The antibody response against both recombinant IrSPI and IrLip1 increased in mice after successive immunizations, and IgG1 antibody titers remained at high levels until day 90 post immunization (Figure 2). Antibody titers were higher for IrLip1 than IrSPI.

Regarding sheep, vaccination with IrSPI, IrLip1, and a combination of IrSPI and IrLip1 all elicited an antigen-specific IgG antibody response that increased after successive immunizations in all groups, peaking at day 45 post injection (Figure 3). Antibody titers within the groups were much less homogeneous than in mice, revealing substantial interindividual variation, but, like in mice were higher against the IrLip1 protein than against the IrSPI protein, whether the proteins were injected alone or in combination. The antibody titers against both IrLip1 and IrSPI were similar in sera from the sheep vaccinated with the proteins alone or with the combination of IrLip1 and IrSPI, thus demonstrating the absence of either synergistic or competitive interactions between the immune response elicited against the two antigens.

### 3.2. Impact of Vaccination on Tick Parameters

In mice, vaccination with IrSPI had no significant impact on any of the evaluated tick parameters, including feeding success (percentage of engorgement and weight after feeding), molting success to adults, or mortality after feeding (Table 1). For mice vaccinated with IrLip1, a small but statistically significant increase was observed in the percentage of engorged ticks and weight after feeding in relation to the control group. Due to the similarity of the antibody titers obtained in the 5 IrLIP1-vaccinated mice, correlation analysis between antibody titers at the time of tick infestation and the impact of vaccination could only be performed for IrSPI-vaccinated mice. The single marginally significant correlation (R^2^ = −0.5) between the level of antibody and the nymph mortality after feeding was negative, suggesting a reduction in tick mortality due to the antibodies produced against IrSPI.

Regarding sheep, the recovery and counting of all non-engorged and engorged larvae made it possible to estimate the actual number of larvae used for infestation and to calculate all of the relevant tick parameters, so as to compare vaccinated and non-vaccinated control sheep. Vaccination, whether with either IrSPI or IrLip1 or both of them combined, was observed to have a significant positive impact on engorgement and molting into nymphs and a significant negative effect on larval mortality (Table 2). By contrast, a positive correlation was found between antibody titers against IrSPI or IrLip1 and tick mortality for sheep vaccinated with IrSPI, IrLip1, or the two antigens, while a negative correlation was found for both larvae engorgement and molting except for anti-IrLip1 antibodies and larval molting in sheep vaccinated with IrLip1 (Appendix A). Nevertheless, with the exception of results obtained for sheep vaccinated with IrSPI alone, the measured correlation depended mainly upon the divergent response of a single individual, and thus, they may not be reliable.

Despite a high overall attachment rate (80%), the global percentage of engorged nymphs that detached by themselves was very low with a mean of 1.5% on the 24 infested sheep. Thus, tick mortality was evaluated according to the number of attached nymphs. Regarding the impact of vaccination on nymph parameters, we observed a significant decrease in tick engorgement for sheep vaccinated with IrSPI and IrLip1 (Table 2). No impact was demonstrated regarding molting efficacy to adults nor tick mortality, except for a significant increase in mortality for ticks fed on IrLip1-vaccinated sheep. However, the total number of engorged and molted nymphs was too low to draw any conclusion about vaccine efficacy for nymphs on vaccinated sheep. This limitation also applies to correlation analysis between antibody titers and tick parameters, which therefore could not be performed.

### 3.3. Impact of Vaccination on A. phagocytophilum Transmission

Following sheep infestation with A. phagocytophilum-infected nymphs, with the exception of a single sheep belonging to the control group, the recorded temperatures remained within normal values, and no bacteria could be detected by PCR (Figure 1). In addition to lethargy and inappetence, this sheep registered an increase in temperature beginning on Day 5 of tick infestation, when the temperature reached 40.5 °C, and a peak of 42 °C, 6 days after infestation (Day 52). The presence of *A. phagocytophilum* in the sheep was evidenced by PCR on blood collected at days 5 and 10 after tick infestation. Upon sequencing of the PCR-amplified msp4 gene, the PCR product displayed 100% identity to the corresponding amplicon of the inoculum, confirming that the animal acquired the bacterium from infected nymphs, and 98.8% identity to the reference sequence of *A. phagocytophilum* strain NV2Os (Gene Bank accession number CP015376.1). It should be noted that this sheep was the one on which the highest number of nymphs became engorged, with a total of 10 engorged nymphs out of 48, versus a general average rate of engorgement of 1.5% (corresponding to 0 to 3 ticks maximum having completed their blood meal on other sheep). Therefore, due to the low level of nymph engorgement on the majority of sheep, we were unable to evaluate the impact of vaccination on *A. phagocytophilum* transmission by ticks.

## 4. Discussion

In order to identify tick factors that play essential roles in either tick biology or pathogen transmission, which may thus represent potential vaccine candidates for tick and TBP control, we used NGS in a previous study to compare gene expression in SG tissue from uninfected and *B. henselae*-infected *I. ricinus* ticks [28]. First, genes predicted to encode secreted proteins were selected, since proteins that interfere with pathogen transmission to the host or with the host response are expected to be secreted into tick saliva and introduced into the host. Moreover, secreted proteins represent attainable targets for neutralization by antibodies elicited by anti-tick vaccines. Following a preliminary functional annotation of this subset, two proteins—IrSPI and IrLip1—were identified as being of particular interest as vaccine candidates and warranting immediate evaluation as such both in mice and sheep. Indeed, to successfully feed and transmit TBD agents, ticks have to overcome serine protease-mediated host defense pathways that are tightly controlled by tick inhibitors, including Kunitz-type inhibitors such as IrSPI [30]. In fact, we have previously demonstrated that *IrSPI* silencing leads to a decrease in the amount of blood ingested by ticks [28], that the native protein is found in tick saliva and is recognized by the serum of tick-infested rabbits, and that its recombinant form is able to decrease T-cell proliferation in mouse splenocytes [31]. Similarly, the inflammatory response is a major host defense pathway that ticks must evade to complete feeding and transmit pathogens [22], and IrLip1, whose RNA is present in tick salivary glands during feeding [28], may similar to other tick lipocalins be presumed to play a role in evasion of the host response through the sequestration of histamine, which is released at the tick-feeding site [45]. Thus, immunization against IrSPI and IrLip1, and more particularly the production of antibodies with the capacity to disrupt their activity, seemed likely to disrupt the acquisition of the blood meal by ticks by limiting the immunosuppressive impact of tick saliva. Nevertheless, the opposite effect was evidenced in the present study, as vaccination was observed to have a positive effect on tick feeding, molting, and viability, for both nymphs on mice and larvae on sheep.

The administration of recombinant IrSPI and IrLip1 was well-tolerated in mice and sheep and elicited an antigen-specific antibody response in both species. As compared with subunit vaccines produced in mammalian or bacterial cells, recombinant proteins expressed in insect cells by baculovirus vectors are expected to adopt a conformation that more closely resembles that of the native protein produced in ticks and may thus promote the induction of antibodies more likely to recognize and neutralize the cognate salivary proteins injected at the site of a tick bite. Both proteins elicited a durable antibody response, although the antibody titers induced by IrLip1 were higher and more homogeneous in both mice and sheep than those induced by IrSPI. The administration of a combination of the two proteins produced the same immunologic profile, suggesting the absence of synergistic or competitive interactions between IrSPI and IrLip1 in the induction of their cognate antibody response. The heterogeneity of the immune response elicited in sheep stood in sharp contrast with the homogeneity observed in mice, underscoring the difference between the murine laboratory model and sheep, and thus the importance of validating the efficacy of anti-tick vaccine candidates in the target species.

Although functional analysis by RNA interference (RNAi) had strongly suggested that IrSPI is implicated in tick feeding [28], comparison between control and vaccinated mice or sheep did not evidence the expected decrease in the engorgement of nymphs or larvae, respectively, in response to vaccination. This may indicate that results obtained by RNAi do not necessarily predict those obtained through vaccination, possibly owing to the off-target effects of RNAi, and it may prove to be misleading in the selection of protective antigens, as has previously been reported [46]. Indeed, statistical comparisons revealed that vaccination against IrSPI had no measurable impact on any of the tick parameters in experiments performed in mice, and it actually appeared to be of benefit to ticks that fed on immunized sheep, since for larvae, both engorgement and molting appeared to be enhanced and mortality diminished. Similar results were obtained with IrLip1 in sheep whether administered alone or together with IrSPI, while a positive effect was also observed on engorgement for nymphs that fed on IrLip1-vaccinated mice. Preliminary evidence of vaccine efficiency for both antigens as regards nymph engorgement on vaccinated sheep with all antigens needs to be reproduced with a greater number of engorged nymphs. Indeed, the capacity of vaccination against tick antigens to protect against tick bites may well vary among animal species, as has already been observed for example between mice and rabbits following immunization with Iris, which is a salivary antigen of *I. ricinus* [24]. Variation in vaccine efficacy between the two species might also be related to the route of vaccine injection, as subcutaneous and intramuscular injection, used here in mice and rabbits, respectively, may have targeted different antigen-presenting cells and elicited distinctly different immune responses. Of note, the intramuscular route was used here in sheep, as it has been shown to be superior to the subcutaneous route for antibody production in this animal species [47]. A difference in fitness due to infection with *A. phagocytophilum* in nymphs used for sheep can also not be excluded.

Several explanations may be advanced for the lack of protection against tick bites following vaccination. First, although for IrSPI we previously ascertained its inoculation with saliva, as evidenced by the production, following tick bites, of IrSPI-specific antibodies [31], we do not know with certainty whether antibodies against IrLip1 are produced following tick infestation. In addition, the functional domains of the proteins, which are not necessarily antigenic, might be unaffected by the generated antibodies, and the targeted epitopes might have no relation to functionality. We also did not evaluate whether ingurgitated antibodies traffic via the tick midgut and hemocoel to the salivary gland, where they could affect the proteins’ function prior release via saliva into immunized animals.

Another possible explanation could be that the functional contribution of these proteins was possibly more modest or different from what was expected under the experimental conditions. In particular, our primary selection of gene candidates was performed on pre-fed *I. ricinus* adult females [28], whereas protein expression is known to vary among the different tick life stages—that is, larvae, nymphs, and adults [48]. Although we previously demonstrated that IrSPI is well expressed at all stages [31], we do not have any data regarding the expression of IrLip1 in larvae and nymphs. Indeed, it would have been interesting to test the impact of vaccination on female engorgement, as they feed over a long period of time, leading to greater exposure to antibodies. It is also known that the protein profile and abundance in tick salivary glands and saliva is dynamic during the course of tick feeding, as has been demonstrated for several tick species [49,50,51,52], and as we have previously confirmed for IrSPI [31]. Thus, it is possible that IrSPI and IrLip1 are not highly expressed during feeding by larvae and nymphs and/or do not make a significant positive contribution to tick engorgement and viability during these stages, which would render them less than ideal as vaccine candidates. In future studies, it would be useful to include early onset and long-term expression during feeding as criteria for the selection of salivary vaccine candidates, as was done by Kim and co-workers for the tick *I. scapularis* [50] and more recently for *Amblyomma americanum* [51]. Such a strategy would permit the selection of proteins that are highly expressed at the beginning of the feeding step, preceding TBP transmission. It has also been demonstrated that the profile of tick salivary proteins during feeding can vary depending on the host [53]. In our previous study, the two proteins were selected on the basis of expression in SG of female ticks that had been engorged through membrane feeding on sheep blood [28], and they may well be differentially expressed when ticks are fed on mice or directly on sheep. Finally, the lack of efficacy of vaccination against tick engorgement, viability, or molting could be the functional redundancy within the two protein families to which they belong [30,33].

Regarding the apparent positive effect of vaccination on indicators of tick development (engorgement, molting, and survival), similar adverse effects of the immune response have been noted in a number of circumstances. Recognized or suspected instances of immune-mediated enhancement have concerned vaccination against viral [54] agents or pre-existing immunity to bacterial [55] and parasitic pathogens [56]. Enhancement has been evidenced as an increased replication of the pathogen, exacerbation of disease manifestation, or even increased susceptibility to infection [57]. In most cases, vaccine-elicited antibodies have been presumed to mediate enhancement, such as by promoting viral attachment to susceptible host cells that display Fc receptors [58], although enhancement has also been observed in the absence of vaccinal antigen-specific antibodies, suggesting that other immune mechanisms can be involved [59,60]. To the best of our knowledge, immune-mediated enhancement has never before been reported after vaccination against an ectoparasite. Rather, the induction of anti-tick immunity has afforded some measure of protection against ticks in multiple reports, and it has been correlated for diverse antigens with an antibody response [61,62,63,64,65] or for the tick cement protein 64TRP with cell-mediated immunity [26,66].

Nevertheless, the potential for immune-mediated enhancement may reside in the nature of the relationship maintained between ticks and the vertebrate immune system. Indeed, *I. ricinus* ticks are exposed to the mammalian immune system for an extended length of time during their blood meals, and in most instances, these hosts will already have encountered tick antigens. The capacity of the tick to elude or resist the host immune response has been extensively documented [22], but it is also conceivable that the tick is able to subvert certain immune responses for its own advantage. Among immune effectors elicited in the present vaccination study, we measured only IrSPI- and IrLip1-specific antibodies. Such antibodies may hypothetically have facilitated the blood meal if, rather than neutralizing the antigens, they actually enhanced their pharmacological activity, such as by increasing the half-life or enhancing attachment to cellular target molecules. Alternatively, and though seemingly unlikely, the presence of IrSPI and IrLip1 in injected saliva might be deleterious in some way to the tick, such that their neutralization by vaccine-elicited antibodies or the neutralization of antigenically related salivary proteins was of benefit to the tick. It is unclear how cell-mediated immune effectors, such as antigen-specific T lymphocytes, could have enhanced blood meal acquisition, but their involvement cannot be formally excluded. It is also possible that vaccination increased susceptibility in an antigen-independent manner, for instance by polarizing the immune response such that the response to tick infestation was less effective. However, this last possibility does not appear likely, as enhancement in vaccinated animals was appreciated in reference to animals inoculated with adjuvant only, and it is the adjuvant component of the vaccine formulation that is presumed to have the greatest impact in determining the T-helper orientation of the vaccinal immune response [67].

## 5. Conclusions

In conclusion, although both recombinant IrSPI and IrLIp1 elicited substantial antigen-specific antibody responses, they failed to afford protection against infestation by *I. ricinus* nymphs and larva in mice and sheep, respectively. Rather, vaccination with either antigen appeared to enhance tick engorgement and molting and decrease tick mortality. While the impact of vaccination on pathogen transmission could not be evaluated in the present study, an enhancement of tick engorgement and survival suggests that it might also facilitate the transfer of pathogens to immunized hosts. Thus, our results underscore that the most critical phase in the development of anti-tick vaccines is the initial screening of protective antigens. Indeed, vaccinal efficacy cannot be inferred with certainty from the functional annotation of tick antigens, and it can only be determined by in vivo trials in the target species. In addition, the question arises of whether the selection of proteins strongly expressed during engorgement, as made here, is judicious. Indeed, perhaps a high expression of salivary antigens is compatible with successful feeding only because they are poorly antigenic, and that less highly expressed salivary antigens might actually represent superior immunogens. Clearly, it is only by pursuing the international effort to identify protective tick antigens that an effective control strategy against these formidable vectors of human and animal diseases will be found. 

## Figures and Tables

**Figure 1 vaccines-08-00475-f001:**
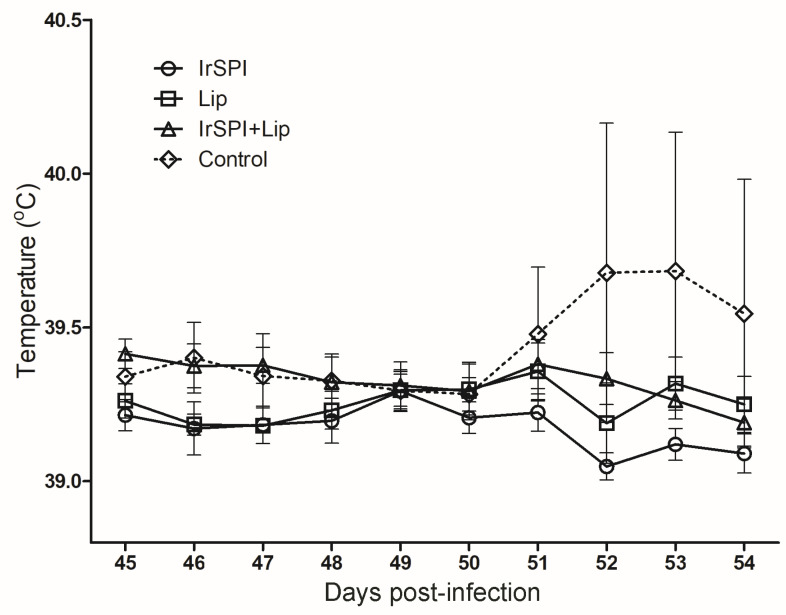
Daily temperature (°C) recorded for groups of sheep vaccinated with IrSPI (*I. ricinus* serine protease inhibitor), IrLip1 (*I. ricinus* lipocalin 1), IrSPI and IrLip1, and the control group injected with adjuvant from 45 to 54 days after immunisation. Animals were infested with An*aplasma phagocytophilum-*infected *Ixodes ricinus* nymphs on day 46. The dotted line representing the control group that received only adjuvant has a very large standard deviation due to a single sheep with a high fever of up to 42 °C on day 52. Results are presented as means ± standard deviation.

**Figure 2 vaccines-08-00475-f002:**
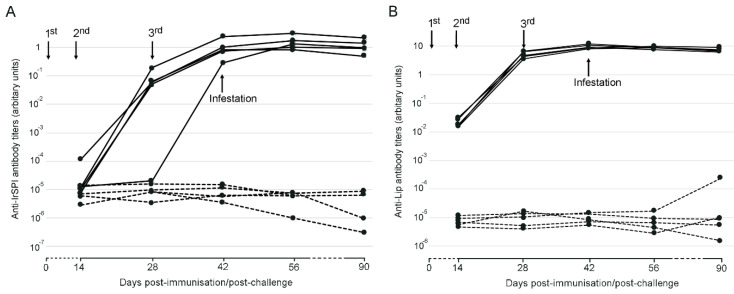
Antibody (immunoglobulin G1, or IgG1) response to recombinant IrSPI (**A**) and IrLip1 (**B**) in vaccinated mice. Antibody titers were determined by ELISA in serum samples collected at different time points from day 0 to day 90 against the specific protein both in vaccinated mice (solid lines) and control mice (dashed lines) that received only adjuvant. Arrows indicate dates for 1st, 2nd, and 3rd immunizations (days 0, 14, and 28) and tick infestations (Day 42). Antibody titers are represented as arbitrary units.

**Figure 3 vaccines-08-00475-f003:**
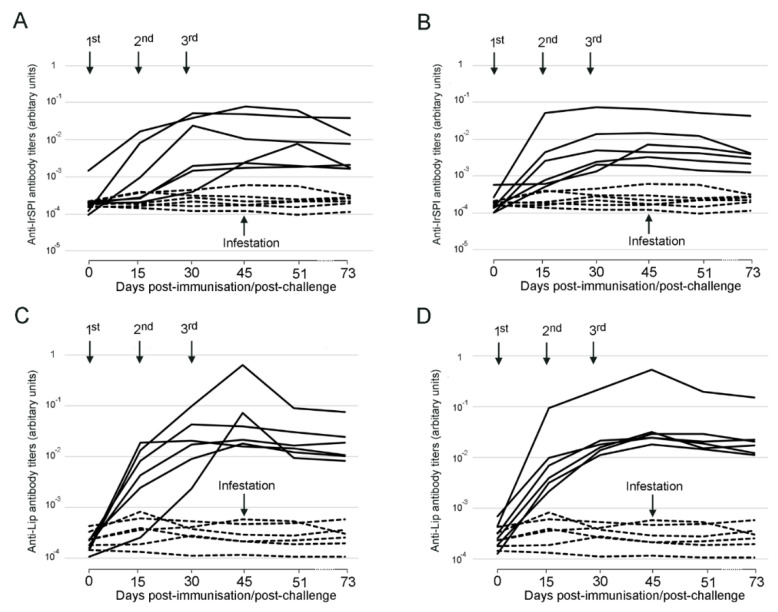
Antibody (immunoglobulin G, or IgG) response to recombinant IrSPI (**A**,**B**) and IrLip1 (**C**,**D**) in sheep vaccinated with recombinant IrSP1 (**A**), recombinant IrLip1 (**C**), or both recombinant IrSPI and Lip (**B**,**D**). Antibody titers were determined by ELISA in serum samples collected at different time points from Day 0 to Day 73 against the specific protein both in vaccinated sheep (solid lines) and control sheep (dashed lines) that received only adjuvant. Arrows indicate dates for 1st, 2nd, and 3rd immunizations (days 0, 15, and 30) and tick infestations (Days 45, 46). Antibody titers are represented as arbitrary units.

**Table 1 vaccines-08-00475-t001:** Effect of vaccination of mice with recombinant IrSPI and IrLip1 proteins on tick infestation parameters.

Tick Parameters	Fed on Control Mice	Fed on IrSPI-Vaccinated Mice	Fed on IrLip1-Vaccinated Mice
Engorgement (number of engorged nymphs/number of nymphs used for infestation)	83/100 (83%)	85/100 (85%)	94/100 (94%) *
Feeding (mean weight after feeding/engorged nymph) (mg)	4.21 (±1.18)	4.22 (±1.18)	4.7 (±1.21) *
Molting (number adults at day 57/number engorged nymphs)	66/83 (79%)	74/85 (87%)	82/94 (87%)
Tick mortality (number of dead nymphs at day 57/total number engorged nymphs)	9/83 (11%)	12/85 (14%)	10/94 (10%)

Results are shown per group of 5 mice analyzed for each condition. Data were analyzed statistically to compare results between nymphs fed on vaccinated and control mice by student’s *t*-test (* *p* < 0.05) and *X*^2^-test (* *p* < 0.05).

**Table 2 vaccines-08-00475-t002:** Effect of vaccination of sheep with recombinant IrSPI and IrLip1 proteins on tick infestation parameters.

Tick Parameters	Fed on Control Sheep	Fed on IrSPI-Vaccinated Sheep	Fed on IrLip1-Vaccinated Sheep	Fed on IrSPI + IrLip1-Vaccinated Sheep
Larvae				
Engorgement (number of engorged larvae/number of larvae used for infestation)	3456/5032 (68%)	4375/5845 (74%) **	5767/6327 (91%) **	4378/5358 (81%) **
Tick mortality (number of dead ticks at Day 90/total number engorged larvae)	2595/3456 (75%)	2316/4375 (53%) **	1871/5767 (32%) **	1161/4378 (26%) **
Molting (number of nymphs at Day 90/number of alive engorged larvae)	138/861 (16%)	400/2059 (19%) *	1112/3896 (28%) **	974/3217 (30%) **
Nymphs				
Engorgement (number of engorged nymphs/number of nymphs used for infestation)	11/288 (4%)	2/288 (0.7%) *	1/288 (0.3%) *	6/288 (2%)
Tick mortality (number of dead ticks at Day 90/total number attached nymphs)	212/219 (96%)	198/200 (99%)	259/260 (99%) *	242/244 (99%)
Molting (number of adults at Day 90/number of engorged nymphs)	7/11 (63%)	2/2 (100%)	1/1 (100%)	2/6 (33%)

Results are shown per group of 6 sheep analyzed for each condition. Data were analyzed statistically to compare results between ticks fed on vaccinated and control sheep by *X*^2^-test (* *p* < 0.05, ** *p* < 0.005).

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
