# Peer review of "Failed Disruption of Tick Feeding, Viability, and Molting after Immunization of Mice and Sheep with Recombinant Ixodes ricinus Salivary Proteins IrSPI and IrLip1"

_vaccines, 2020, doi:10.3390/vaccines8030475_

Round 1

Reviewer 1 Report

This manuscript by Almazan, et al. describes the vaccination of mice and sheep with two tick salivary proteins with the intention of impeding tick feeding by the immune response generated. The rationale for choosing these proteins is that they are up-regulated upon Bartonella acquisition and serve to likely enhance feeding and transmission. While the report essentially conveys “negative” data, it does provide valuable insight into the process of identifying candidate anti-tick vaccines. The emphasis appears to be on enhancement of tick feeding by vaccination, yet it is not clear that this is justified by the number of animals/ticks tested and possible significance. The difference in sheep is more pronounced than in mice. It would be better to change the emphasis to “failed disruption” of tick feeding by vaccination with these 2 proteins.

-Suggest title change to; “Failed disruption (or similar wording) of tick feeding, viability and molting after immunization with..”

-In the results section, the mouse antibody responses were measured and reported for IgG1. Why were other isotypes not considered?

-The results are clearly presented and proper statistical tests were performed.

-To the discussion, multiple ideas may be added.

                The question of whether or not the antibodies generated bind and “neutralize” the proteins remains unanswered. It could be that the functional domains are unaffected and not antigenic, i.e. that targeted epitopes have no effect on functionality.

                Another question is whether or not antibodies to these proteins are naturally produced following tick feeding.

                Given that these proteins are highly expressed may suggest that the hosts have evolved to not generate good antibody responses to ensure adequate tick feeding. Perhaps it would be better to select subdominant antigens for immunization.

                The numerous factors in tick saliva which quell the local immune response must be overcome in order for any anti-tick vaccine to work. A discussion of how tick saliva may impair responses even in the presence of specific antibody is warranted.

                Redundancy in function is mentioned as a plausible explanation for the results; this is very likely for the lipocalin, for which there exist many paralogues.

Author Response

This manuscript by Almazan, et al. describes the vaccination of mice and sheep with two tick salivary proteins with the intention of impeding tick feeding by the immune response generated. The rationale for choosing these proteins is that they are up-regulated upon Bartonella acquisition and serve to likely enhance feeding and transmission. While the report essentially conveys “negative” data, it does provide valuable insight into the process of identifying candidate anti-tick vaccines. The emphasis appears to be on enhancement of tick feeding by vaccination, yet it is not clear that this is justified by the number of animals/ticks tested and possible significance. The difference in sheep is more pronounced than in mice. It would be better to change the emphasis to “failed disruption” of tick feeding by vaccination with these 2 proteins.

Many thanks for your positive comments. Please find below, response to the specific comments.

-Suggest title change to; “Failed disruption (or similar wording) of tick feeding, viability and molting after immunization with..”

Thanks for this suggestion, the title was modified as requested: “Failed disruption of tick feeding, viability and molting after immunization of mice and sheep with recombinant Ixodes ricinus salivary proteins IrSPI and IrLip1”

-In the results section, the mouse antibody responses were measured and reported for IgG1. Why were other isotypes not considered?

We agree with the reviewer that the other isotypes could have been analyzed, which we can no longer do now. But, at the time, the important immune response in IgG1 seemed to us sufficient to begin experiments on sheep.

-The results are clearly presented and proper statistical tests were performed.

Thanks for that

-To the discussion, multiple ideas may be added.

The discussion was modified based on the constructive comments of the reviewer and we hope that the new version is more satisfactory

                The question of whether or not the antibodies generated bind and “neutralize” the proteins remains unanswered. It could be that the functional domains are unaffected and not antigenic, i.e. that targeted epitopes have no effect on functionality.

We agree with the reviewer’s comment and the discussion was modify in that sense. In addition, we did not evaluate the passage way of acquired antibodies via tick midgut and haemocoel to salivary gland where they supposed to encounter the natural function of the proteins after feeding on an immunized animal, what is now also discussed in the new version: “In addition, the functional domains of the proteins, which are not necessarily antigenic, might be unaffected by the generated antibodies, and the targeted epitopes might have no relation to functionality. We also did not evaluate whether ingurgitated antibodies traffic via tick midgut and haemocoel to the salivary gland, where they could affect the proteins’ function prior release via saliva into immunized animals..”

                Another question is whether or not antibodies to these proteins are naturally produced following tick feeding.

As mentioned in the introduction “in a recent study, we also showed that IrSPI is a Kunitz elastase inhibitor that is overexpressed in several tick organs—especially SG—during blood feeding, and that is injected into the host through saliva.” Such an injection through the saliva was demonstrated through the production of Abs against IrSPI following tick bites. This information also appears in the discussion “that the native protein is found in tick saliva and is recognized by the serum of tick-infested rabbits, »

However, no such experiment was done for IrLip1 and we do not know if some Abs against this protein are produced following tick bites. The discussion was modified in the new version in order to clarify that point: “Several explanations may be advanced for the lack of protection against tick bites following vaccination. First, although for IrSPI we previously ascertained its inoculation with saliva, as evidenced by the production, following tick bites, of IrSPI-specific antibodies [31], we do not know with certainty whether antibodies against IrLip1 are produced following tick infestation. “

                Given that these proteins are highly expressed may suggest that the hosts have evolved to not generate good antibody responses to ensure adequate tick feeding. Perhaps it would be better to select subdominant antigens for immunization.

This is a very good suggestion, thanks a lot for that, and the idea was added in the discussion section :” In addition, the question arises of whether the selection of proteins strongly expressed during gorgement, as made here, is judicious. Indeed, perhaps high expression of salivary antigens is compatible with successful feeding only because they are poorly antigenic, and that less highly expressed salivary antigens might actually represent superior immunogens.”

                The numerous factors in tick saliva which quell the local immune response must be overcome in order for any anti-tick vaccine to work. A discussion of how tick saliva may impair responses even in the presence of specific antibody is warranted.

We agree with the reviewer’s comment. However, we think that a discussion on the topic would be more appropriate for a general review on anti-tick vaccination than in a research paper like this one. However, the following sentence was added to the introduction in response to this relevant remark: “However, vaccine-elicited antibodies against at least some tick antigens have proven to afford protection against natural challenge, despite possible interference by immunosuppressive activities of tick saliva. Encouraging results have been actually obtained for several tick species … »

                Redundancy in function is mentioned as a plausible explanation for the results; this is very likely for the lipocalin, for which there exist many paralogues.

yes, this is indeed true

Reviewer 2 Report

This is a very interesting paper and shows that even the best hypothesis may be shown to be incorrect after proper experimental testing. 

The work done is well performed and described. I would be inclined to exclude the work you did on  transmission of Anaplasma phagocytophilum by nymphs to sheep as this work couldn't actually be done satisfactorily due to technical difficulties. It clutters the paper, which is otherwise clear in its results on the vaccinations.

The title need to be a bit clearer by adding the following "...molting after immunization of mice and sheep with recombinant...."

The tick vaccine based on the antigen Bm86 was also used commercially in Australia in 1994. Not sure if it still is however.

This paper is the classic case of a "great" vaccine in theory, failing miserably when tested experimentally. It actually enhanced tick engorement and molting and decreased tick mortality !

The paper is quite long and the transmission of Anaplasma phagocytophilum component of the paper could be omitted as it wasn't able to be  done  properly.

Author Response

This is a very interesting paper and shows that even the best hypothesis may be shown to be incorrect after proper experimental testing. 

The work done is well performed and described. I would be inclined to exclude the work you did on transmission of Anaplasma phagocytophilum by nymphs to sheep as this work couldn't actually be done satisfactorily due to technical difficulties. It clutters the paper, which is otherwise clear in its results on the vaccinations.

Many thanks for your positive comments. Please find below, response to the specific comments.

The title need to be a bit clearer by adding the following "...molting after immunization of mice and sheep with recombinant...."

As requested by the reviewers 1 and 2 the title was modified as follow in the new version: “Failed disruption of tick feeding, viability and molting after immunization of mice and sheep with recombinant Ixodes ricinus salivary proteins IrSPI and IrLip1”

The tick vaccine based on the antigen Bm86 was also used commercially in Australia in 1994. Not sure if it still is however.

The vaccine based on Bm86 is no more commercialized in Australia, that’s the reason why we did not mention it.

This paper is the classic case of a "great" vaccine in theory, failing miserably when tested experimentally. It actually enhanced tick engorement and molting and decreased tick mortality !

The paper is quite long and the transmission of Anaplasma phagocytophilum component of the paper could be omitted as it wasn't able to be done properly.

We understand the reviewer’s point of view. However, as we know that the infection of ticks can modify their fitness -what was specify in the new version-, we cannot omit that the nymphs used in sheep were infected with Anaplasma phagocytophilum. In the new version of the manuscript we have therefore tried to reduce this part but it seems to us complicated to eliminate it completely despite the bad results obtained.

Reviewer 3 Report

The manuscript by Almazán and colleagues describes a vaccine trial where mice and sheep mounted an antibody response to recombinant tick salivary peptides, but the ticks subsequently fed on them were minimally affected, or seemed to do better than ticks fed on controls. Attempts to challenge the sheep with A. phagocytophilum by tick bite were unsuccessful, likely due to the use of infected nymphs that fed very poorly on the sheep. The authors argue that expression of the chosen serpin and lipocalin in immature ticks was unknown, and could have been low, which might explain why ticks were not affected. It’s not clear why the authors did not determine levels of expression for the respective genes in the ticks used. It seems the intent was to use infected adults for challenge of vaccinated sheep, but this was not possible for technical reasons (no infected adult ticks were obtained), and the feeding rate of larvae and nymphs on the sheep was extremely low. One sheep in the control group became infected, but the sequence of the genes evaluated did not match the challenge strain, casting doubt on its origin. While it is valuable to report negative results, in this case, the results cannot be explained by the data, because of the poor feeding success of the ticks, which was essentially a failure, rendering the tick infestation and challenge meaningless. This part of the study should be removed or summarized in a few sentences.

Abstract

I suggest removing the mention of the failed A. phagocytophilum experiment, as it does not add anything of scientific importance, given there were not enough infected nymphs.

Line 27: larvae instead of larva

Line 28: To the best of our knowledge

Line 45: I. ricinus is also a vector for Babesia microti that infects humans

There is some redundancy in the Introduction, e.g., lines 37-40, 48-49 and 53-54

Line 75-76: The hosts of R. microplus include a number of domestic and wild animals, not just cattle

Lines 126-128: here and in the remainder of the manuscript, the effect of vaccination on A. phagocytophilum transmission by ticks should be cut back to a brief summary. This aspect of the research failed for reasons unrelated to the vaccine trial, and this information is thus not relevant to the present study, and no conclusions can be drawn. 

Lines 147 and on: While S2 cells may be a convenient and easy to use system for expressing recombinant tick proteins, they are also unlikely to provide the appropriate post-translational modifications (not the same in ticks and insects!), which might explain the negative results.

Line 177: perhaps better “in the abdominal region” to avoid confusion with i.p. injection 

There are some sections in the Methods where text is italicized, but should not be.

Line 193: Platform? I think you mean facility or institute?

Line 197: Why were sheep injected intramuscularly while mice were injected subcutaneously for immunization? These different sites target different immune cell populations, please discuss, because there will be differences in the immune response elicited due to the route of administration.

Line 199: “cotton cells” is not a term that is familiar to those working with ticks. Looking back at the cited references, the term used is “stockinet cells,” which is not any more clear. “Stockinet” is a brand name for cotton fabric tubing used by patients suffering from edema. The term “stockinette” refers to cotton fabric tubing, which is commonly used for confining ticks on animals. 

Line 200: Each sheep was. Line 201 “that had engorged as larvae.” “Engorged” is not used in the same way as “fed”

Line 234: please state the source of the ISE6 cells, e.g., the Tick Cell Biobank, or other

Lines 289-291: please rewrite, I do not understand what you mean to say here

Table 1 and others: please place a period after “No” so that it is clear it means “number” instead of “no” Also, the tables should list the actual tick numbers, not just some proportion. It must be completely clear how many ticks were used for the infestations.

Line 314: I suggest you replace global with overall. 

Line 314 – 315: I. ricinus nymphs do not naturally feed on sheep, which probably explains the low rate of feeding success on sheep

Line 319: molting to adults

Line 331: why 98% identity? Which gene was this? Msp4 of 16S rRNA?

Lines 396 and on: The Methods and the Results section indicate that infected nymphs were used, not adult females, so this statement is confusing.

The discussion aiming to explain why vaccination enhanced tick feeding and survival is highly speculative. If it was not known how much the target proteins were expressed in immature ticks, their choice was unfortunate, but this could (and should) be verified to lend a more factual basis to the discussion.

In the Supplementary table, decimal values should be indicated using periods, not commata. Also, the legend should provide more detail.

Author Response

The manuscript by Almazán and colleagues describes a vaccine trial where mice and sheep mounted an antibody response to recombinant tick salivary peptides, but the ticks subsequently fed on them were minimally affected, or seemed to do better than ticks fed on controls. Attempts to challenge the sheep with A. phagocytophilum by tick bite were unsuccessful, likely due to the use of infected nymphs that fed very poorly on the sheep. The authors argue that expression of the chosen serpin and lipocalin in immature ticks was unknown, and could have been low, which might explain why ticks were not affected. It’s not clear why the authors did not determine levels of expression for the respective genes in the ticks used.

As mention in the MS, it has been done for IrSPI (“Although we previously demonstrated that IrSPI is well expressed at all stages [31] …» but effectively not for IrLIP1 and we agree that this should have been done.

It seems the intent was to use infected adults for challenge of vaccinated sheep, but this was not possible for technical reasons (no infected adult ticks were obtained), and the feeding rate of larvae and nymphs on the sheep was extremely low.

That is true for the nymphs on sheep, not for the larvae for which it varies from 68% to 92% depending of the group.

 One sheep in the control group became infected, but the sequence of the genes evaluated did not match the challenge strain, casting doubt on its origin.

We do not understand that comment as it is mention in the MS that “The amplified sequence from the positive sheep showed 98 % identity with A. phagocytophilum NV2Os, which confirmed that the animal acquired the infection by infected nymphs”. For us, taking into consideration possible problem due to amplification and sequencing, this % if enough to clam that this is the same strain that to one used to infect ticks. The real % of identity was 98.8% what was added in the new version.

While it is valuable to report negative results, in this case, the results cannot be explained by the data, because of the poor feeding success of the ticks, which was essentially a failure, rendering the tick infestation and challenge meaningless. This part of the study should be removed or summarized in a few sentences.

This part was then modified as required.

Abstract

I suggest removing the mention of the failed A. phagocytophilum experiment, as it does not add anything of scientific importance, given there were not enough infected nymphs.

We understand the reviewer’s point of view. However, as we know that the infection of ticks can modify their fitness, we cannot omit that the nymphs used in sheep were infected with Anaplasma phagocytophilum. In the new version of the manuscript we have therefore tried to reduce this part but it seems to us complicated to eliminate it completely despite the bad results obtained.

Line 27: larvae instead of larva

This was corrected

Line 28: To the best of our knowledge

This was corrected

Line 45: I. ricinus is also a vector for Babesia microti that infects humans

Although some DNA of B. microti can be found in I. ricinus, and one experimental study demonstrated the possible transmission of the parasite to gerbils, there is no epidemiological evidence of transmission by this tick species in Europe to date. From our knowledge, in Europe, B. microti is transmitted to rodents by I. trianguliceps, which do not feed on humans, explaining why, contrary to North American, there is no verified cases of human disease due to this parasite in Europe. Ixodes ricinus biting very frequently humans, if this tick being a vector of Babesia microti, there would certainly be human cases in Europe. In this continent, human cases of Babesiosis are mostly due to B. divergens and it does not correspond to a real public health problem, justifying why babesiosis does not appear in our list for human diseases.

There is some redundancy in the Introduction, e.g., lines 37-40, 48-49 and 53-54

Initially, the beginning of the paragraph referred to vector borne diseases in general when the end of the paragraph referred specifically to ticks borne diseases. however, the paragraph has been amended as the reviewer sees redundancy, and to deal directly with ticks.

Line 75-76: The hosts of R. microplus include a number of domestic and wild animals, not just cattle

We agree that it is possible to find some R. microplus on some horses, goats, sheep, donkeys, dogs, pigs or some wild mammals, however, as mentioned in the manuscript, we maintain that this species of tick, called the cattle tick, feeds principally on cattle and that other livestock species or wild ungulates are rarely parasitized: “…the targeted tick species feeds principally on the host for which the vaccine is intended. While such holds true for the so-called cattle tick R. microplus…”

Lines 126-128: here and in the remainder of the manuscript, the effect of vaccination on A. phagocytophilum transmission by ticks should be cut back to a brief summary. This aspect of the research failed for reasons unrelated to the vaccine trial, and this information is thus not relevant to the present study, and no conclusions can be drawn. 

This part was reduced in the new version of the MS and we hope that it is now better

Lines 147 and on: While S2 cells may be a convenient and easy to use system for expressing recombinant tick proteins, they are also unlikely to provide the appropriate post-translational modifications (not the same in ticks and insects!), which might explain the negative results.

We agree that post-translational modification can be different between ticks and insects but we think that using arthropod cells is probably closer to the native proteins than using bacterial system for example. We have previously demonstrated (Blisnick et al 2019) that the antibodies generated against the recombinant protein IrSPI are able to recognize the native protein but we did not demonstrate it for IrLIP1. However, it is clear that in addition, as mentioned by the reviewer 1, the functional domains could be unaffected and not antigenic, i.e. that targeted epitopes have no effect on functionality and that point was added in the new version.

Line 177: perhaps better “in the abdominal region” to avoid confusion with i.p. injection 

This was corrected

There are some sections in the Methods where text is italicized, but should not be.

This was corrected

Line 193: Platform? I think you mean facility or institute?

This was corrected

Line 197: Why were sheep injected intramuscularly while mice were injected subcutaneously for immunization? These different sites target different immune cell populations, please discuss, because there will be differences in the immune response elicited due to the route of administration.

We agree that it might change the immune cells targeted but we followed the “classical” immunization route used for each animal species. In mice, the SC route is the most useful way of injection for practical reasons. In sheep, both ways are recommended, and the IM normally doesn’t cause any secondary effects when appropriately done. The immune response also depends of the adjuvant and the one used here (Montanide™ ISA 201 VG, Seppic), has been proved to enhance immune response when intramuscularly applied in cattle. In addition, recently it was demonstrated that in sheep the IM route is better that SC in terms of antibody production (van Rijn et al. 2017).  

Accordingly, the following sentences were added in the discussion section of the new version: “Variation in vaccine efficacy between the two species might also be related to the route of vaccine injection, as subcutaneous and intramuscular injection, used here in mice and rabbits, respectively, may have targeted different antigen presenting cells and elicited distinctly different immune responses. Of note, the intramuscular route was used here in sheep as it has been shown to be superior to the subcutaneous route for antibody production in this animal species [46]. »

Line 199: “cotton cells” is not a term that is familiar to those working with ticks. Looking back at the cited references, the term used is “stockinet cells,” which is not any more clear. “Stockinet” is a brand name for cotton fabric tubing used by patients suffering from edema. The term “stockinette” refers to cotton fabric tubing, which is commonly used for confining ticks on animals. 

 “Cotton cells” has been extensively used to describe tick infestation experiments. See the following references for example: Almazán C, Lagunes R, Villar M, et al. Identification and characterization of Rhipicephalus (Boophilus) microplus candidate protective antigens for the control of cattle tick infestations. Parasitol Res. 2010;106(2):471-479. doi:10.1007/s00436-009-1689-1 ; Canales M, Almazán C, Naranjo V, Jongejan F, de la Fuente J. Vaccination with recombinant Boophilus annulatus Bm86 ortholog protein, Ba86, protects cattle against B. annulatus and B. microplus infestations. BMC Biotechnol. 2009;9:29. Published 2009 Mar 31. doi:10.1186/1472-6750-9-29….

However, according to the reviewer’s comment the term “cotton cells” was replaced by “cotton bags “on the new version as it is a simple cotton fabric here and not “stockinette”

Line 200: Each sheep was. Line 201 “that had engorged as larvae.” “Engorged” is not used in the same way as “fed”

This was corrected

Line 234: please state the source of the ISE6 cells, e.g., the Tick Cell Biobank, or other

As required all this part regarding A. phagocytophilum was reduced and now the materials and methods refer to our previous paper

Lines 289-291: please rewrite, I do not understand what you mean to say here

This sentence was rewritten and we hope that it is clearest in the new version

Table 1 and others: please place a period after “No” so that it is clear it means “number” instead of “no” Also, the tables should list the actual tick numbers, not just some proportion. It must be completely clear how many ticks were used for the infestations.

This was modified

Line 314: I suggest you replace global with overall. 

This was corrected

Line 314 – 315: I. ricinus nymphs do not naturally feed on sheep, which probably explains the low rate of feeding success on sheep

We are sorry but we do understand that comment because many studies show that the nymphs of Ixodes ricinus can feed on sheep in the field. Among others, see for example the paper of Ogden and co-workers: OGDEN, N., CASEY, A., FRENCH, N., ADAMS, J., & WOLDEHIWET, Z. (2002). Field evidence for density-dependent facilitation amongst Ixodes ricinus ticks feeding on sheep. Parasitology, 124(2), 117-125.

Line 319: molting to adults

This was corrected

Line 331: why 98% identity? Which gene was this? Msp4 of 16S rRNA?

Sorry for this mistake that was corrected, the amplified gene was MSP4 and the percentage of identity being equal to 98.8%, we estimate that it well corresponds to the bacterial strain used to infect ticks and transmitted to the sheep by them.

Lines 396 and on: The Methods and the Results section indicate that infected nymphs were used, not adult females, so this statement is confusing.

This part was deleted;

The discussion aiming to explain why vaccination enhanced tick feeding and survival is highly speculative. If it was not known how much the target proteins were expressed in immature ticks, their choice was unfortunate, but this could (and should) be verified to lend a more factual basis to the discussion.

We believe it is possible to be speculative in a discussion. In addition, the effect observed following vaccination represents the first report of induction of vaccine-mediated enhancement in relation to anti-tick vaccination so it seems to us interesting to discuss it. As mentioned above, the expression was validated in immature stages for IrSPI. Therefore, this explanation could only be valid for IrLIP1 and the discussion was reviewed to make it clearer.

In the Supplementary table, decimal values should be indicated using periods, not commata. Also, the legend should provide more detail.

All the figures have been revised, comma replaced by periods, and the quality improved

Round 2

Reviewer 3 Report

The revised manuscript is improved, but authors’ responses to some of my suggestions are not satisfactory, see below. For example, the fact that larvae fed well on the sheep is irrelevant, since Anaplasma is not transmitted transovarially. If nymphs that fed as larvae on sheep did not feed again, nothing was accomplished with respect to transmission. Also, the assertion that the strain detected in the one infected control sheep was the same as the challenge strain is erroneous. The sequences should be identical, and amplicons should have been sequenced in both directions. This is not the same as a case where a field isolate was compared to data bank entries to obtain an identification.

Line 39: According to the European Centers for Disease Control, I. ricinus is a vector of Babesia species that infect humans in Europe. Therefore, leaving human babesiosis out of your list is not justified. At the ECDC website (https://www.ecdc.europa.eu/en/all-topics-z/babesiosis/facts-about-babesiosis), the following statement appears: “Human transmission of Babesia species in Europe is mainly due to Ixodes ricinus. The distribution and frequency of this tick implies that human babesiosis could occur in any part of Europe. The 39 published human cases in Europe were clinically severe and were attributed to B. divergensB. venatorum (EU1) and B. microti.” And on the same page: “In Europe, human transmission occurs mainly by an Ixodes ricinus tick bite.”

The statement of what hosts cattle fever ticks feed on is wrong. Apparently, the authors are unaware of the recognized importance of wildlife in the maintenance of this tick species, as well as of pathogens that can infect both wildlife and cattle. These are not isolated incidences, and have been recognized for many years. Examples are the maintenance of R. microplus and annulatus by exotic antelope and deer in Texas (George et al. 1989. WILDLIFE AS A CONSTRAINT TO THE ERADICATION OF BOOPHILUS SPP. (ACARI: IXODIDAE) J. Agric. Entomol. 7(2): 119-125; Busch et al. Widespread movement of invasive cattle fever ticks (Rhipicephalus microplus) in southern Texas leads to shared local infestations on cattle and deer. Parasites Vectors 7, 188 (2014); Lohmeyer et al. Implication of Nilgai Antelope (Artiodactyla: Bovidae) in Reinfestations of Rhipicephalus (Boophilus) microplus (Acari: Ixodidae) in South Texas: A Review and Update. J Med Entomol. 2018 May 4;55(3):515-522 Foley et al. Prev Vet Med. Movement patterns of nilgai antelope in South Texas: Implications for cattle fever tick management 2017 Oct 1;146:166-172), as well as adaptation of R. microplus to deer in New Caledonia (De Meeûs et al. Swift sympatric adaptation of a species of cattle tick to a new deer host in New Caledonia. Infect Genet Evol. 2010 Oct;10(7):976-83.). This is a problem that is not new, and has grown dramatically in recent years, facilitating expansion of the cattle tick that can no longer be considered to be under control.

Concerning he discussion – yes, it is possible to be speculative, but it is not necessarily appropriate. In the case of this manuscript, given the problems with experimental design, it is not appropriate. There is no known reason for the failed nymphal feeding on sheep.

Author Response

The revised manuscript is improved, but authors’ responses to some of my suggestions are not satisfactory, see below.

We are sorry that the reviewer has not been satisfied with our replies to certain of his/her suggestions and shall attempt to provide more satisfactory responses.

For example, the fact that larvae fed well on the sheep is irrelevant, since Anaplasma is not transmitted transovarially.

We believe that the reviewer is making reference to the following exchange from the first round of reviewing:

[Reviewer # 3] It seems the intent was to use infected adults for challenge of vaccinated sheep, but this was not possible for technical reasons (no infected adult ticks were obtained), and the feeding rate of larvae and nymphs on the sheep was extremely low.

[Authors] That is true for the nymphs on sheep, not for the larvae for which it varies from 68% to 92% depending of the group

The reviewer has correctly understood that we originally wished to evaluate the impact of vaccination not only on tick parameters but also on transmission of A. phagocytophilum. The selected strategy was to infect ticks at the larval stage during feeding on an infected sheep, to allow the larvae to molt into nymphs, and then to infest vaccinated sheep with infected nymphs as the virulent challenge. Most unfortunately, the % of engorged nymphs was very low, thus precluding assessment of transmission and restricting evaluation of the impact of vaccination to its effect on tick parameters.  While we do not know why engorgement of nymphs was so low, the problem appeared to be tick stage-specific, since the feeding rate of larvae on sheep was satisfactory (68-92% depending on the group).

If nymphs that fed as larvae on sheep did not feed again, nothing was accomplished with respect to transmission.

We agree with the reviewer that owing to this problem we were unable to evaluate transmission, and that evaluation of the impact of vaccination was restricted to its effect on tick parameters.

Also, the assertion that the strain detected in the one infected control sheep was the same as the challenge strain is erroneous. The sequences should be identical, and amplicons should have been sequenced in both directions. This is not the same as a case where a field isolate was compared to data bank entries to obtain an identification.

The amplicons were in fact sequenced in both directions. We are sorry for having misspoken, as in fact the sequencing results revealed 100% identity with the inoculum (cell culture supernatant) used to infect the sheep and hence ticks (thus confirming that the sheep acquired the bacteria via our infected ticks).  We did, however, wish to make reference to the original field strain deposited in genbank, and that actually displays 98.8% identity with the cultured strain, presumably owing to genetic selection or drift during culture. To avoid any confusion in the MS we propose:

 "Upon sequencing of the PCR-amplified msp4 gene, the PCR product displayed 100 % identity to the corresponding amplicon of the inoculum, confirming that the animal acquired the bacterium from infected nymphs, and 98.8 % identity to the reference sequence of A. phagocytophilum strain NV2Os (Gene Bank accession number CP015376.1). "

Line 39: According to the European Centers for Disease Control, I. ricinus is a vector of Babesia species that infect humans in Europe. Therefore, leaving human babesiosis out of your list is not justified. At the ECDC website (https://www.ecdc.europa.eu/en/all-topics-z/babesiosis/facts-about-babesiosis), the following statement appears: “Human transmission of Babesia species in Europe is mainly due to Ixodes ricinus. The distribution and frequency of this tick implies that human babesiosis could occur in any part of Europe. The 39 published human cases in Europe were clinically severe and were attributed to B. divergensB. venatorum (EU1) and B. microti.” And on the same page: “In Europe, human transmission occurs mainly by an Ixodes ricinus tick bite.”

We completely agree with the reviewer that I ricinus has been implicated in transmission of Babesia to humans in Europe. Nevertheless, given the rarity of such cases we did not spontaneously include human babesiosis in our list of medically important diseases transmitted by I ricinus. However, at the request of the reviewer, human babesiosis was included in the new version.

The statement of what hosts cattle fever ticks feed on is wrong. Apparently, the authors are unaware of the recognized importance of wildlife in the maintenance of this tick species, as well as of pathogens that can infect both wildlife and cattle. These are not isolated incidences, and have been recognized for many years. Examples are the maintenance of R. microplus and annulatus by exotic antelope and deer in Texas (George et al. 1989. WILDLIFE AS A CONSTRAINT TO THE ERADICATION OF BOOPHILUS SPP. (ACARI: IXODIDAE) J. Agric. Entomol. 7(2): 119-125; Busch et al. Widespread movement of invasive cattle fever ticks (Rhipicephalus microplus) in southern Texas leads to shared local infestations on cattle and deer. Parasites Vectors 7, 188 (2014); Lohmeyer et al. Implication of Nilgai Antelope (Artiodactyla: Bovidae) in Reinfestations of Rhipicephalus (Boophilus) microplus (Acari: Ixodidae) in South Texas: A Review and Update. J Med Entomol. 2018 May 4;55(3):515-522 Foley et al. Prev Vet Med. Movement patterns of nilgai antelope in South Texas: Implications for cattle fever tick management 2017 Oct 1;146:166-172), as well as adaptation of R. microplus to deer in New Caledonia (De Meeûs et al. Swift sympatric adaptation of a species of cattle tick to a new deer host in New Caledonia. Infect Genet Evol. 2010 Oct;10(7):976-83.). This is a problem that is not new, and has grown dramatically in recent years, facilitating expansion of the cattle tick that can no longer be considered to be under control.

We fully agree with the reviewer that R microplus feeds on hosts other than cattle, but had felt justified in stating that cattle represent a “preferential” host for this tick. In the revised version we have nuanced and clarified our argument by describing cattle simply as a “major” host : “For vaccines such as these, however, whose impact on TBD is secondary to their effect on tick burden, reduction in TBD is unlikely to be achieved unless the targeted tick species feeds principally on the host for which the vaccine is intended. Such holds true for the so-called cattle tick R. microplus, which while capable of feeding on a variety of hosts mainly infests cattle in areas where they are present. Such is not, however, the case for many species of ticks responsible for important TBD, and notably for Ixodes sp., which feed relatively indiscriminately on multiple hosts and notably on wildlife. For these ticks, vaccines that interrupt tick feeding prior to pathogen transmission or that directly suppress vector competence must be sought. »

Concerning the discussion – yes, it is possible to be speculative, but it is not necessarily appropriate. In the case of this manuscript, given the problems with experimental design, it is not appropriate. There is no known reason for the failed nymphal feeding on sheep.

In the first round of reviewing, the reviewer stated that the “discussion aiming to explain why vaccination enhanced tick feeding and survival is highly speculative”. Let us just say that such enhancement was completely unexpected and, to the best of our knowledge, represents the first description in the literature as regards anti-tick vaccines. We recognize that the possible explanations that we provided are highly speculative, but any explanation of a newly described phenomenon would be, or so it seems to us.  

The reviewer evokes “problems with experimental design”, by which we presume he/she refers to the poor feeding rate of infected nymphs on sheep. If so, we should like to reassure the reviewer that this technical problem would have had no impact on evaluation of the impact of vaccination on nymphs fed on mice, nor on larvae fed on sheep, in which cases the feeding rate was perfectly adequate. Since vaccine-mediated enhancement was potentially observed in these cases, our “speculative” explanations refer to data that we see as being robust.

Round 3

Reviewer 3 Report

The present version satisfies my criticisms.